# Searching the Search Space of Vision Transformer

**Minghao Chen[1,*], Kan Wu[2,*], Bolin Ni[3,*], Houwen Peng[4,†],**
**Bei Liu[4], Jianlong Fu[4], Hongyang Chao[2], Haibin Ling[1]**
[1]Stony Brook University, [2]Sun Yat-sen University,
[3]Institute of Automation, CAS, [4]Microsoft Research

## Abstract

Vision Transformer has shown great visual representation power in substantial vision tasks such as recognition and detection, and thus been attracting fast-growing efforts on manually designing more effective architectures. In this paper, we propose to use neural architecture search to automate this process, by searching not only the architecture but also the search space. The central idea is to gradually evolve different search dimensions guided by their *E-T Error* computed using a weight-sharing supernet. Moreover, we provide design guidelines of general vision transformers with extensive analysis according to the space searching process, which could promote the understanding of vision transformer. Remarkably, the searched models, named S3 (short for *Searching the Search Space*), from the searched space achieve superior performance to recently proposed models, such as Swin, DeiT and ViT, when evaluated on ImageNet. The effectiveness of S3 is also illustrated on object detection, semantic segmentation and visual question answering, demonstrating its generality to downstream vision and vision-language tasks. Code and models will be available at here.

## 1 Introduction

Vision transformer recently has drawn great attention in computer vision because of its high model capability and superior potentials in capturing long-range dependencies. Building on top of transformers [41], modern state-of-the-art models, such as ViT [10] and DeiT [40], are able to achieve competitive performance on image classification and downstream vision tasks [13, 20] compared to *CNN* models. There are fast-growing efforts on manually designing more effective architectures to further unveil the power of vision transformer [27, 53, 15].

Neural architecture search (NAS) as a powerful technique for automating the design of networks has shown its superiority to manual design [58, 39]. For NAS methods, search space is crucial because it determines the performance bounds of architectures during the search. It has been observed that the improvement of search space induced a favorable impact on the performance of many state-of-the-art models [2, 16, 50]. Researchers have devoted ample efforts to the design of *CNN* space [45, 12, 42]. However, limited attention is paid to *Transformer* counterparts, leading to the dilemma of search space design when finding more effective and efficient transformer architectures.

In this paper, we propose to Search the Search Space (S3) by automatically refining the main changeable dimensions of transformer architectures. In particular, we try to answer the following two key questions: *1) How to effectively and efficiently measure a specific search space? 2) How to automatically change a defective search space to a good one without human prior knowledge?*

For the first question, we propose a new measurement called *E-T Error* for search space quality estimation and utilize a once-for-all supernet for effective and efficient computing. Specifically, the "E" part in the E-T error is similar to the error empirical distribution function used in [35] focusing on

---

*Equal contributions. Work done when Minghao, Kan, Bolin are interns of MSRA. †Corresponding author.

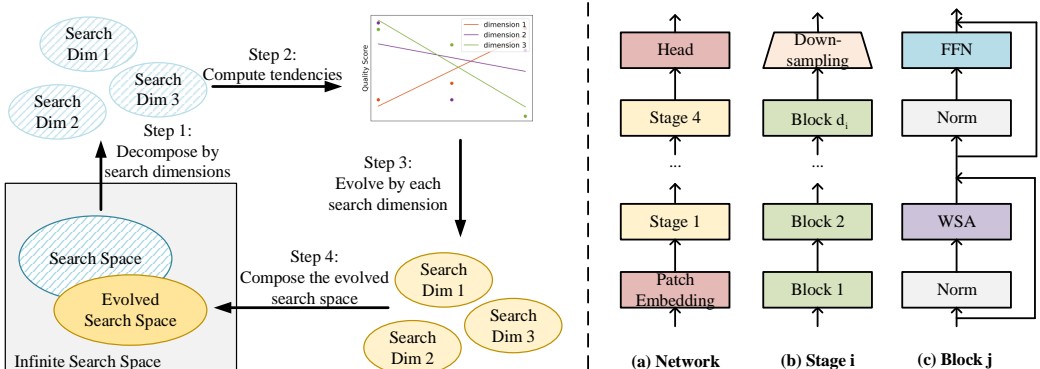

Figure 1: **Left:** The pipeline of searching the search space. **Right:** The general vision transformer architecture explored in this paper.

the overall quality, while the "T" part considers the quality of top-tier architectures in a search space. In contrast to [35] that trains each subnet for only a few epochs, we train a once-for-all supernet with AutoFormer [6] to obtain a large amount of well-trained subnets, providing a more reliable proxy for performance estimation.

For the second question, as shown in Fig. 1 we decompose the search space by multiple dimensions, including depth, embedding dimension, MLP ratio, window size, number of heads and $Q$-$K$-$V$ dimension, and progressively evolve each dimension to compose a better space. Particularly, we use linear parameterization to model the tendencies of different dimensions so as to guide the search.

In addition, we analyze the search process of search space and provide some interesting observations and design guidelines: 1) The third stage is the most important one and increasing its number of blocks leads to performance improvement. 2) Shallow layers should use a small window size while deep layers should use a large window size. 3) MLP ratio is better to be progressively increased along with network depth. 4) $Q$-$K$-$V$ dimension could be smaller than the embedding dimension without performance drop. We hope these observations will help both the manual design of vision transformer and the space design for automatic search.

In summary, we make the following contributions:

- We propose a new scheme, *i.e.* S3, to automate the design of search space for vision transformer. We also present a new pipeline for vision transformer search with minimal human interactions. Moreover, we provide analysis and guidelines on vision transformer architecture design, which might be helpful for future architecture search and design.
- The experiments on ImageNet verify the proposed automatic search space design method can improve the effectiveness of design space, thus boosting the performance of searched architectures. The discovered models achieve superior performance compared to the recent ViT [10] and Swin [27] transformer families under aligned settings. Moreover, the experiments on downstream vision and vision-language tasks, such as object detection, semantic segmentation and visual question answering, demonstrate the generality of the method.

## 2 Approach

### 2.1 Problem Formulation

Most of existing NAS methods can be formulated as a constrained optimization problem as:

$$\alpha_{\mathcal{A}}^* = \arg\min_{\alpha \in \mathcal{A}} \mathcal{L}(W_\alpha^*; \mathcal{D}_{\text{val}}) \quad s.t. \quad W_\alpha^* = \arg\min_{W_\alpha} \mathcal{L}(W_\alpha; \mathcal{D}_{\text{train}}), \quad g(\alpha) < c, \tag{1}$$

where $W_\alpha$ is the weights of the network architecture $\alpha$, $\mathcal{A}$ is a predefined search space, $\mathcal{L}(\cdot)$ is the loss function, $\mathcal{D}_{\text{train}}$ and $\mathcal{D}_{\text{val}}$ represent train and validation datasets respectively, $g(\cdot)$ denotes a function calculating resource consumption of a model while $c$ denotes a given resource constraint.

Ideally, the search space $\mathcal{A}$ is an infinite space $\Omega$ that contains all possible architectures. In practice, $\mathcal{A}$ is usually a small subspace of $\Omega$ due to limited computation resource. In this paper, we break the

convention of searching in a fixed space by considering the architecture search together with the search space design. More concretely, we decouple the constrained optimization problem of Eq. (1) into three separate steps:

**Step 1.** Searching for an optimal search space under a specific constraint:

$$\mathcal{A}^* = \min_{\mathcal{A} \subset \Omega} \mathcal{Q}(\mathcal{A}; \mathcal{D}_{\text{val}}) \quad s.t. \ |\mathcal{A}| \leq M, \tag{2}$$

where $\mathcal{Q}$ is a measurement of search space to be elaborated in Sec. 2.3, and $M$ represents the maximum search space size.

**Step 2.** Following one-shot NAS [12, 34], we encode the search space into a supernet and optimize its weights. This is formulated as solving the following problem:

$$W^*_{\mathcal{A}^*} = \arg\min_{W} \mathbb{E}_{\alpha \in \mathcal{A}^*}[\mathcal{L}(W_\alpha; \mathcal{D}_{\text{train}})], \tag{3}$$

where $W$ is the weights of the supernet, $W_\alpha$ is the weights in $W$ specified by the architecture $\alpha$.

**Step 3.** After obtaining the well-trained supernet, we search for the optimal architecture via ranking the performance using the learned weights, which is formulated as:

$$\alpha^*_{\mathcal{A}^*} = \arg\min_{\alpha \in \mathcal{A}^*} \mathcal{L}(W^*_\alpha; \mathcal{D}_{\text{val}}), \ \ s.t. \ g(\alpha) < c, \tag{4}$$

where $W^*_\alpha$ is the weights of architecture $\alpha$ inherited from $W^*_{\mathcal{A}^*}$.

## 2.2 Basic Search Space

We set up the general architectures of vision transformer by following the settings in ViT [10] and Swin-T [27] as shown in Fig. 1 (right). To be specific, given an input image, we uniformly split it into non-overlap patches. Those patches are linearly projected to vectors named patch embeddings. The embeddings are then fed into the transformer encoder, which performs most of the computation. At last, a fully connected layer is adopted for classification.

The transformer encoder consists of four sequential stages with progressively reduced input resolutions. Each stage contains blocks with the same embedding dimension. Therefore, stage $i$ has two search dimensions: the number of blocks $d_i$ and embedding dimension $h_i$. Each block contains a window-based multi-head self-attention (WSA) module and a feed-forward network (FFN) module. We do not force the blocks in one stage to be identical. Instead, block $j$ in the $i^{th}$ stage have several search dimensions including window size $w^i_j$, number of heads $n^i_j$, MLP ratio $m_i$, Q-K-V embedding dimension $q^i_j$.

## 2.3 Searching the Search Space

*Space Quality Evaluation.* For a given space $\mathcal{A}$, we propose to use *E-T Error*, denoted as $\mathcal{Q}(\mathcal{A})$, for evaluating its quality. It is the average of two parts: expected error rate $\mathcal{Q}_e(\mathcal{A})$ and top-tier error $\mathcal{Q}_t(\mathcal{A})$. For a given space $\mathcal{A}$, the definition of $\mathcal{Q}_e(\mathcal{A})$ is given as:

$$\mathcal{Q}_e(\mathcal{A}) = \mathbb{E}_{\alpha \in \mathcal{A}, g(\alpha) < c}[e_\alpha], \tag{5}$$

where $e_\alpha$ is the top-1 error rate of architecture $\alpha$ evaluated on ImageNet validation set, $g(\cdot)$ is the computation cost function and $c$ is the computation resource constraint. It measures the overall quality of a search space. In practice, we use the average error rate of $N$ randomly sampled architectures to approximate the expectation term. The top-tier error $\mathcal{Q}_t(\mathcal{A})$ is the average error rate of top 50 candidates under resource constraints, representing the performance upper bound of the search space.

*Once-for-all Supernet.* In constrast to RegNet [35] that trains hundreds of models with only a few epochs ($\sim$10) and uses their errors as a performance proxy of the fully trained networks, we encode the search space into a supernet and optimize it similar to AutoFormer [6]. Under such training protocol, we are able to train an once-for-all transformer supernet that serves as a reliable and efficient performance indicator.

*Searching the Search Space.* The search of search space mainly contains two iterative steps: 1) supernet optimization and 2) space evolution.

**Algorithm 1** Searching the Search Space

---

**Input:** Infinite search space $\Omega$, max iteration $T$, threshold $\tau$ for subspace evolved, the subspaces $S_1^{(0)}, \cdots, S_D^{(0)}$, evolve steps for subspaces $\gamma_1, \cdots, \gamma_D$
**Output:** The most promising search space $\mathcal{A}^*$

1: Initialize a search space $\mathcal{A}^{(0)}$ from $\Omega$ with minimal prior human knowledge
2: **while** $t \in [0, 1, \cdots, T]$ **do**
3:      Optimize the weights $W_{\mathcal{A}}^{(t)}$ of supernet corresponding to the space $\mathcal{A}^{(t)}$
4:      Randomly sample $N$ architectures from the well converged supernet
5:      **for** space $S_i^{(t)}$ in $S_1^{(t)}, \cdots, S_D^{(t)}$ **do**
6:          Get the subspace $\{\alpha | \alpha \in \mathcal{A}^{(t)} \cap \alpha_i = v_l^{(t)}\}$ for each choice $v_l^{(t)} \in S_i^{(t)}$, where $\alpha_i$ is the $i^{th}$ dimension value of the architecture $\alpha$
7:          Calculate the *E-T Error* of each subspace according to Eq. (5)
8:          Evolve the subspace and get $S_{D_i}^{(t+1)}$ according to Eq. (8)
9:      **end for**
10:      $\mathcal{A}^* = \mathcal{A}^{(t+1)} = S_1^{(t+1)} \times S_2^{(t+1)} \times \cdots \times S_D^{(t+1)}$
11: **end while**

---

**Step 1.** For the search space $\mathcal{A}^{(t)}$ of $t^{th}$ iteration, we encode it into a supernet. We adopt sandwich training [52] to optimize the weights of supernet, *i.e.*, sampling the largest, the smallest, and two middle models for each iteration and fusing their gradients to update the supernet. Notably, we do not use inplace-distill since it leads to collapse for transformer training.

**Step 2.** We first decompose the search space $\mathcal{A}^{(t)}$ by the search dimensions, defined in Sec .2.2, and four stages. Denote their corresponding subspaces as $S_1^{(t)}, S_2^{(t)}, \cdots, S_D^{(t)}$, where $D$ is the product of the number of stage and the number of search dimensions. The search space $\mathcal{A}^{(t)}$ now can be expressed as the Cartesian product of all search spaces:

$$\mathcal{A}^{(t)} = S_1^{(t)} \times S_2^{(t)} \times \cdots \times S_D^{(t)}, \tag{6}$$

Then the search space $\mathcal{A}^{(t)}$ could be evolved to a better one $\mathcal{A}^{(t+1)}$, with the update of $S_i^{(t)}$. Particularly, For a certain subspace $S_i^{(t)}$, we first get the subspace $\{\alpha | \alpha \in \mathcal{A}^{(t)} \cap \alpha_i = v_l^{(t)}\}$ for each choice $v_l^{(t)} \in S_i^{(t)}$, where $\alpha_i$ is the $i^{th}$ dimension value of the architecture $\alpha$. Then we compute the E-T Errors of the subspaces. We hypothesize there is a strong correlation between the continuous choices and the E-T Error in most cases, so we fit a linear function to estimate its tendency for simplification:

$$\mathbf{y} = w\mathbf{x} + b, \tag{7}$$

where $\mathbf{x} \in \mathbb{R}$ is the choices in $S_i^{(t)}$, and $\mathbf{y} \in \mathbb{R}$ is the corresponding E-T Error. $w$ and $b$ are both scalar parameters. Then the new subspace $S_i^{(t+1)}$ is given as:

$$S_i^{(t+1)} = \left\{ v_j^{(t)} - \left\lfloor \frac{w}{\tau} \right\rfloor \gamma_i \ \middle| \ v_j^{(t)} \in S_i^{(t)} \right\}, \tag{8}$$

where $\tau$ is the threshold for search space evolving, $\gamma_i$ is the steps for search dimension $S_i$. Note that we remove the choices if they become less than zero. The detailed algorithm is also shown in Alg. 1.

## 2.4   Searching in the Searched Space

Once the space search is done, we perform neural architecture search over the searched space. The search pipeline contains two sequential steps: 1) supernet training without resource constraints, and 2) evolution search under resource constraints.

The *supernet training* follows the same train recipe in space search. Similar to AutoFormer [6] and BigNas [52], the maximal, minimal subnets and several random architectures are sampled randomly from the evolved search space in each iteration and their corresponding weights are updated. We provide more details in Sec. 4.1. The *evolution search* process follows the similar setting in SPOS [12]. Our objective in this paper is to maximize the classification accuracy while minimizing the model size and FLOPs. Notably, vision transformer does not include any Batch Normalization [19],

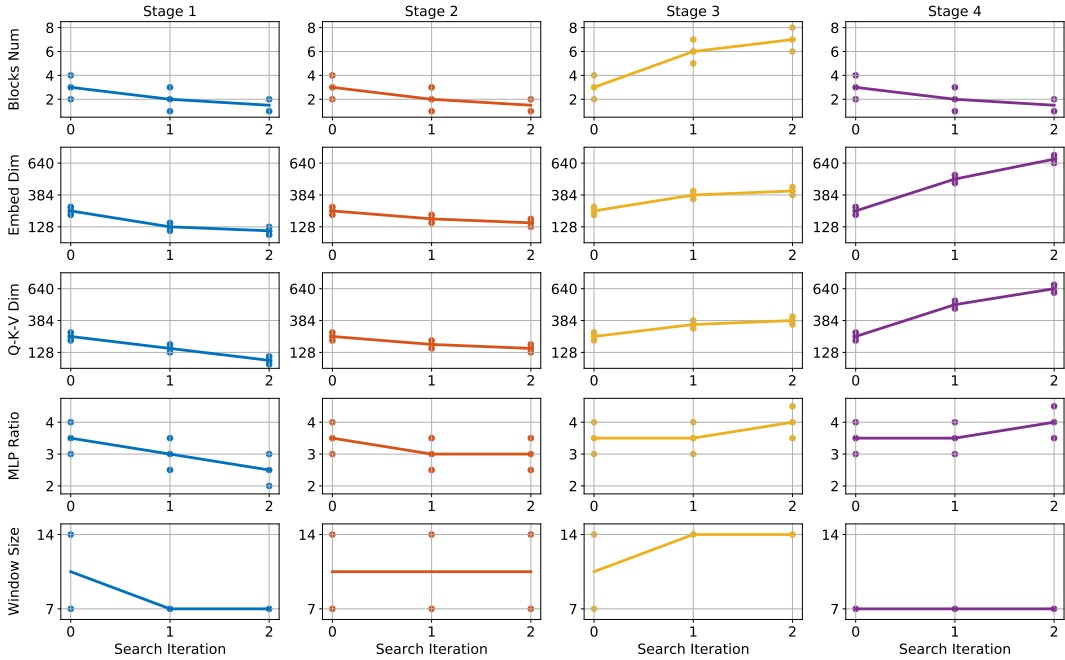

Figure 2: Visualization of space search process. Columns and rows represent different stages and search dimensions. The dots in each sub-figure means the choices of that search dimension. We plot addition lines for better viewing the change tendency of dimensions.

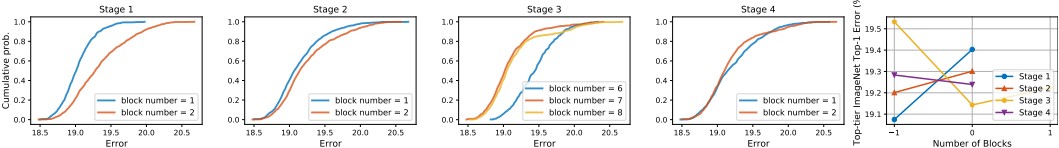

Figure 3: The left 4 subplots present the empirical distribution function of different stages of the searched space focusing on the number of blocks. The rightest is the top-tier error of different stages.

hence the evolution process is much faster than CNN. We provide more details in the supplementary materials.

# 3 Analysis and Discussion

In this section we provide analysis and discussion according to the space searching process. We hope that our observations can shed lights on both manual design of vision transformer architecture and the search space design for NAS methods. To minimize the prior knowledge used in search space design, we initialize the search space by setting a same search space for different stages. We provide the detailed search space and its evolution process in Fig. 2. Fig. 3 shows a direct illustration of space quality of the searched space focusing on the number of blocks.

*The third stage is most important and increasing blocks in the third stage leads to better performance.* As shown in the first row of Fig. 2, the blocks of stage 3 is constantly increasing from step 0 to step 2. We also observe that there are more blocks in stage 3 in some famous CNN networks, such as ResNet [14], EfficientNet [39], which share the same architecture characteristics with our finding in transformers. As a result, it seems that the third stage is the most important one. On the other hand, contrary to stage 3, the number of blocks decrease in stage 1,2,4 showing that these stages should include fewer layers.

*Shallow layers should use a small window size while deep layers should use a larger one.* As shown in the last row of Fig. 2, with the search steps increasing, the window size space decreases in stage 1 while increases in stage 3. It shows that shallow layers tend to have smaller window size while deep layers tend to have larger one. We assume that transformers increase focusing region with deeper layers. To verify this, we visualize the attention maps (top row in Fig. 4 (b,c) ) in different layers and

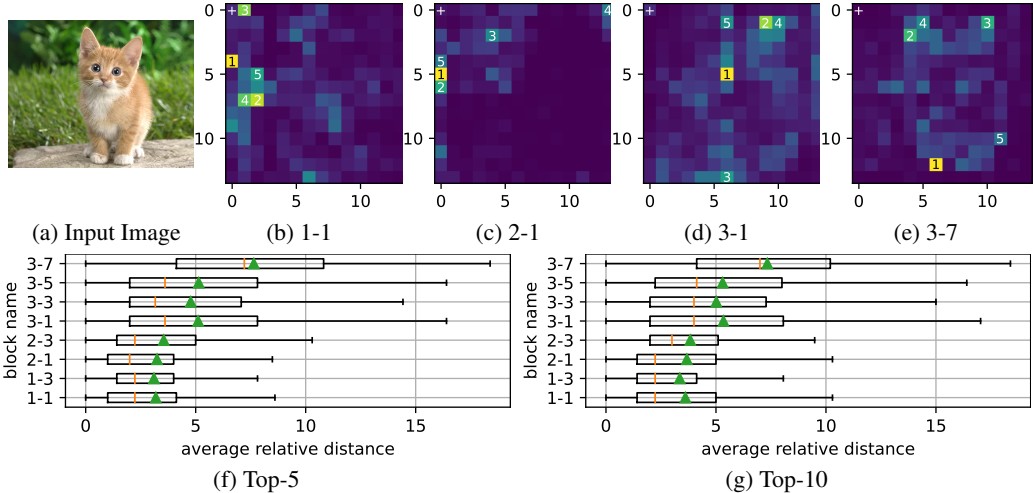

|   |   |   |   |   |
|---|---|---|---|---|
| (a) Input Image | (b) 1-1 | (c) 2-1 | (d) 3-1 | (e) 3-7 |

|   |   |
|---|---|
| (f) Top-5 | (g) Top-10 |

Figure 4: Illustration of attention maps of some selected layers ((b)-(e)), and the average relative distance of a query and its top-k keys with highest attention ((f), (g)). For attention maps ("3-7" represents the "stage 3 - block 7", while (c)-(e) follow the rule), the reference position is on left top, marked with $+$. The numbers 1-5 on attention map subpatches represent the top-5 highest attention. Besides, bottom row shows top-5/10 average relative distances of some selected layers. The orange lines ǀ represent the median and the green triangles ▲ represent the mean.

draw a box plot (bottom row in Fig. 4 (f,g)) to show the relative distances from the target position to its top-5/10 focusing positions. The attention maps show that the highest attentions (noted by numbers) expand with deeper layer, which is illustrated from (b) to (c). As for the average relative distance, deeper layers have larger medians and means, indicating larger focusing regions. The above observations are consistent with our assumption and provide solid evidence for it.

*Deeper layer should use larger MLP ratio.* Traditionally, all layers use the same MLP ratio. However, the third row of Fig. 2 illustrates that shallow layers should use small MLP ratio to result in better performance while deeper ones should use larger MLP ratio.

*Q-K-V dimension should be smaller than the embedding dimension.* In the original transformer block [41, 10], the $Q$-$K$-$V$ dimension is the same as the embedding dimension. However, comparing the second row with the third row of Fig. 2, we find that the searched space has a slightly smaller $Q$-$K$-$V$ dimension than embedding dimension, shown from stage 1 to stage 4. The gap between them are more obvious in the deeper layers. We conjecture the reason is that there are many heads with similar features in deeper layers. Therefore, the a relative small $Q$-$K$-$V$ dimension is able to have great visual representation ability. We present more details in the supplementary materials.

## 4 Experiments

### 4.1 Implementation Details

*Space searching.* The initial search space is presented in Fig. 2, the number of blocks and embedding dimension are set to $\{2, 3, 4\}$ and $\{224, 256, 288\}$ respectively for all stages, while the window size, number of heads, MLP ratio, $Q$-$K$-$V$ dimensions are set to $\{7, 14\}$, $\{3, 3.5, 4\}$, $\{7, 8, 9\}$, $\{224, 256, 288\}$ respectively for all blocks. We set the total iteration of space searching to 3, while setting the steps in Eq. (8) for block number, embed dim, MLP ratio, Q-K-V dim, windows size to 1, 64, 0.5, 64, 7, respectively. We set the threshold $\tau$ of space evolution in Eq. (8) as 0.4.

*Supernet training.* Similar to DeiT [40], we train the supernet with the following settings: AdamW [28] optimizer with weight decay 0.05, initial learning rate $1 \times 10^{-3}$ and minimal learning rate $1 \times 10^{-5}$ with cosine scheduler, 20 epochs warmup, batch size of 1024, 0.1 label smoothing, and stochastic depth with drop rate 0.2. We also adopt sandwich strategy as describe in Sec. 2.3. The models are trained for 300 epochs with 16 Nvidia Tesla 32G V100 GPUs.

Table 1: S3 performance on ImageNet in comparison with state-of-the-arts. We group the models according to their parameter sizes.

| Models | Top-1 Acc. | Top-5 Acc. | #Params. | FLOPs | Res. | Model Type |
|---|---|---|---|---|---|---|
| ResNet50 [14] | 76.2% | 92.9% | 25.5M | 4.1G | $224^2$ | CNN |
| RegNetY-4GF [35] | 80.0% | - | 21.4M | 4.0G | $224^2$ | CNN |
| EfficietNet-B4 [39] | 82.9% | 95.7% | 19.3M | 4.2G | $380^2$ | CNN |
| T2T-ViT-14 [53] | 80.7% | - | 21.5M | 6.1G | $224^2$ | Transformer |
| DeiT-S [40] | 79.9% | 95.0% | 22.1M | 4.7G | $224^2$ | Transformer |
| ViT-S/16 [10] | 78.8% | - | 22.1M | 4.7G | $224^2$ | Transformer |
| Swin-T [27] | 81.3% | - | 28.3M | 4.5G | $224^2$ | Transformer |
| **S3-T (Ours)** | **82.1%** | **95.8%** | **28.1M** | **4.7G** | $224^2$ | Transformer |
| ResNet152 [14] | 78.3% | 94.1% | 60M | 11G | $224^2$ | CNN |
| EfficietNet-B7 [39] | 84.3% | 97.0% | 66M | 37G | $600^2$ | CNN |
| BoTNet-S1-110 [38] | 82.8% | 96.4% | 55M | 10.9G | $224^2$ | CNN + Trans |
| T2T-ViT-24 [53] | 82.2% | - | 64M | 15G | $224^2$ | Transformer |
| Swin-S [27] | 83.0% | - | 50M | 8.7G | $224^2$ | Transformer |
| **S3-S (Ours)** | **83.7%** | **96.4%** | **50M** | **9.5G** | $224^2$ | Transformer |
| **S3-S-384 (Ours)** | **84.5%** | **96.8%** | **50M** | **33G** | $384^2$ | Transformer |
| RegNetY-16GF [35] | 80.4% | - | 83M | 16G | $224^2$ | CNN |
| ViT-B/16 [10] | 79.7% | - | 86M | 18G | $224^2$ | Transformer |
| DeiT-B [40] | 81.8% | 95.6% | 86M | 18G | $224^2$ | Transformer |
| Swin-B [27] | 83.3% | - | 88M | 15.4G | $224^2$ | Transformer |
| Swin-B-384 [27] | 84.2% | - | 88M | 47G | $384^2$ | Transformer |
| **S3-B (Ours)** | **84.0%** | **96.6%** | **71M** | **13.6G** | $224^2$ | Transformer |
| **S3-B-384 (Ours)** | **84.7%** | **96.9%** | **71M** | **46G** | $384^2$ | Transformer |

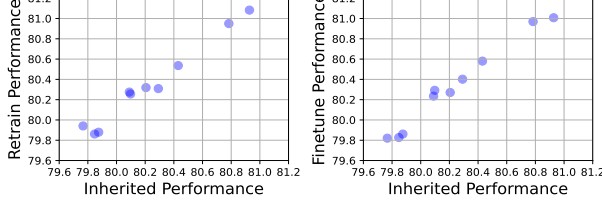

Figure 5: Comparison of subnets with inherited weights, fine-tuned and trained from scratch.

Figure 6: Error empirical distribution function of different search spaces.

## 4.2 Ablation Study

*Effectiveness of once-for-all supernet sampling*. To verify the effectiveness, we randomly sample 10 architectures from the well-trained supernet with the inherited weights, then compare the ImageNet classification performance of (1) models with only inherited weights, (2) models with finetuning, and (3) models retrained from scratch. As shown in Fig. 5, the architecture with inherited weights has comparable or even better performance when compared with the same architecture with retraining or finetuning. This phenomenon shows that our sampled architectures provide a precise and reliable performance proxy for space quality measurement.

*Effectiveness of search space evolution*. We conduct two experiments. In the first experiment, we randomly sample $1,000$ architectures with inherited weights from a search space, then compute and plot error empirical distribution, as in [35], of them. We denote the initial space, the one-step evolved space and the two-step evolved space by Space A, Space B and Space C respectively. As shown in Fig. 6, the space C has an obviously better overall quality compared to A and B. In the second experiment, we use evolution algorithm to search for top architectures in each space. Tab. 5 illustrates that the space C has a much better top architectures, being 16.6%/4.1% better than that in the spaces A and B. These results verify the effectiveness of search spaces evolution.

Table 2: Results on COCO object detection using Cascaded Mask R-CNN. We provide two settings with FPN (1×: 12 epochs; 3×: 36 epochs). Results are taken from [44] and its official repository.

| Backbone | #Params (M) | #FLOPs (G) | Cas. Mask R-CNN w/ FPN 1× | | | | | | Cas. Mask R-CNN w/ FPN 3× | | | | | |
|---|---|---|---|---|---|---|---|---|---|---|---|---|---|---|
| | | | $AP^b$ | $AP^b_{50}$ | $AP^b_{75}$ | $AP^m$ | $AP^m_{50}$ | $AP^m_{75}$ | $AP^b$ | $AP^b_{50}$ | $AP^b_{75}$ | $AP^m$ | $AP^m_{50}$ | $AP^m_{75}$ |
| ResNet-50 [14] | 82 | 739 | 41.2 | 59.4 | 45.0 | 35.9 | 56.6 | 38.4 | 46.3 | 64.3 | 50.5 | 40.1 | 61.7 | 43.4 |
| DeiT-S [44] | 80 | 889 | - | - | - | - | - | - | 48.0 | 67.2 | 51.7 | 41.4 | 64.2 | 44.3 |
| Swin-T [27] | 86 | 745 | 48.1 | 67.1 | 52.2 | 41.7 | 64.4 | 45.0 | 50.4 | 69.2 | 54.7 | 43.7 | 66.6 | 47.3 |
| **S3-T(Ours)** | 86 | 748 | **48.4** | **67.8** | **52.3** | **42.0** | **64.8** | **45.2** | **50.5** | **69.4** | **54.9** | **43.8** | **66.8** | **47.5** |
| ResNeXt-101-32 [48] | 101 | 819 | 44.3 | 62.7 | 48.4 | 38.3 | 59.7 | 41.2 | 48.1 | 66.5 | 52.4 | 41.6 | 63.9 | 45.2 |
| Swin-S [27] | 107 | 838 | 50.3 | 69.6 | 54.8 | 43.4 | 66.7 | 47.0 | 51.8 | 70.4 | 56.3 | 44.7 | 67.9 | 48.5 |
| **S3-S(Ours)** | 107 | 855 | **50.8** | **70.1** | **55.0** | **43.9** | **67.5** | **47.2** | **52.6** | **71.5** | **56.9** | **45.5** | **68.8** | **49.4** |
| ResNeXt-101-64 [48] | 140 | 972 | 45.3 | 63.9 | 49.6 | 39.2 | 61.1 | 42.2 | 48.3 | 66.4 | 52.3 | 41.7 | 64.0 | 45.1 |
| Swin-B [27] | 145 | 982 | 50.5 | 69.5 | 55.0 | 43.5 | 66.9 | 46.9 | 51.9 | 70.7 | 56.3 | 45.0 | 68.2 | 48.8 |
| **S3-B(Ours)** | 128 | 947 | **50.8** | **69.7** | **55.2** | **43.7** | **67.1** | **47.0** | **52.5** | **71.2** | **57.1** | **45.4** | **68.8** | **49.1** |

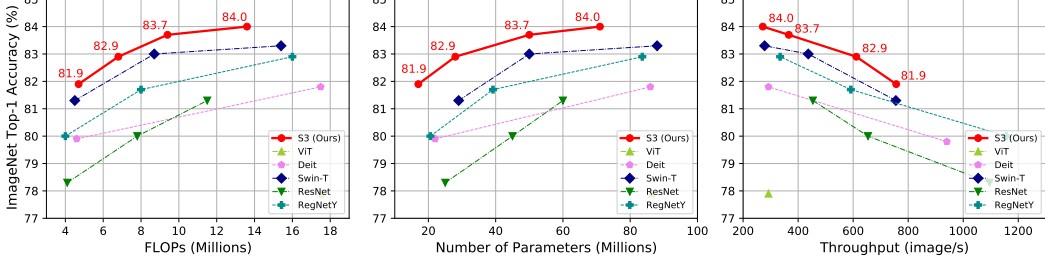

Figure 7: Comparison with state-of-the-art models under different constraints.

## 4.3 Image Classification

We compare our searched models with state-of-the-arts on ImageNet [9]. As described in Sec. 2.1 and Sec. 4.1, we first evolve the search space, then optimize the weights of supernet and finally search on the supernet. For the experiments of high resolution images, the models are finetuned with $384^2$ inputs for 10 epochs, using an AdamW [28] optimizer with a constant learning rate of $10^{-5}$ and weight decay of $10^{-8}$.

We compare our S3 with state-of-the-art CNN models, vision transformers and hybrid CNN-transformer models. The results are reported in Tab. 1. Our proposed models S3-tiny/small/base (T/S/B for short) achieve 82.1%/83.7%/84.0% top-1 accuracy on ImageNet [9] using $224^2$ input images. Compared with vision transformer and hybrid models, our method obtains superior performance using similar FLOPs and parameters. For example, when considering parameters as the main constraint, we find a architecture that has 0.6% higher top-1 accuracy than Swin Transformer [27] while using 42% fewer parameters as shown in Fig. 7 . Compared to CNN models, our S3 models are also very competitive, especially for small parameter size. The proposed S3-S-384 (84.5%) performs slightly better than EfficientNet-B7 (84.3%) [39] while using 24% fewer parameters and 11% fewer FLOPs. For base models with fewer computational cost, S3-B also achieves superior performance to state-of-the-art methods, such as RegNet [35] and ViT [10].

## 4.4 Vision and Vision-Language Downstream Tasks

To further evaluate the generalizability of the discovered architectures, we transfer them to vision and vision-language downstream tasks. We conduct experiments on COCO [24] for object detection, ADE20K [56] for semantic segmentation and VQA 2.0 [54] for visual question answering, respectively. The training and inference settings for object detection and semantic segmentation are the same as those in Swin transformer [27], while for visual language answering task, the settings are aligned with SOHO [18]. More experimental details are elaborated in supplementary materials.

**Object Detection.** As shown in Tab. 2, S3-T outperforms ResNet-50 [14] by +4.2 box AP and +3.7 mask AP under 3× learning schedule. Compared to DeiT-S [40], S3-T achieves +2.5 higher box AP and +2.4 higher mask AP. Increasing the model size and FLOPs, S3-S and S3-B surpass ResNeXt-101 [48] by +4.5 box AP, +3.9 mask AP and +4.2 box AP, +3.7 mask AP respectively. Compared to Swin-S and Swin-B, S3-S and S3-B outperform them by +0.8 box AP, +0.8 mask AP and +0.6 box AP, +0.4 mask AP respectively. All comparisons are conducted under the similar model

Table 3: Results on ADE20K Semantic Segmentation using UperNet. †: reproduced by the official code. ∗: without adding deconvolution in heads.

| Method | Backbone | #param (M) | FLOPs (G) | mIoU (%) | mIoU (ms+flip %) |
|---|---|---|---|---|---|
| UperNet | ResNet-50 [14] | 67 | 951 | 42.05 | 42.78 |
| UperNet | DeiT-S* [40] | 52 | 1099 | 43.15 | 43.85 |
| UperNet | Swin-T [27] | 60 | 945 | 44.51 | 45.81 |
| UperNet | **S3-T(Ours)** | 60 | 954 | **44.87** | **46.27** |
| UperNet | ResNet-101 [14] | 86 | 1029 | 43.82 | 44.85 |
| UperNet | DeiT-B* [40] | 121 | 2772 | 44.09 | 45.68 |
| UperNet | Swin-S† [27] | 81 | 1038 | 47.60 | 48.87 |
| UperNet | **S3-S(Ours)** | 81 | 1071 | **48.04** | **49.31** |

Table 4: Results on VQA v2.0 dataset.

| Backbone | #Param (M) | FLOPs (G) | VQA test-dev(%) | VQA test-std(%) |
|---|---|---|---|---|
| ResNet-50 [14] | 4.1 | 26 | 64.65 | 65.12 |
| ResNet-101 [14] | 7.8 | 45 | 65.44 | 65.71 |
| Swin-T [27] | 4.5 | 29 | 67.13 | 67.40 |
| **S3-T (Ours)** | 4.6 | 29 | **67.84** | **68.00** |

Table 5: Comparisons of searched architectures in different spaces.

| Space | #Param(M) | FLOPs(G) | Top-1(%) |
|---|---|---|---|
| A | 28M | 4.7 | 65.5 |
| B | 29M | 4.7 | 78.0 |
| C | 28M | 4.7 | 82.1 |

size and FLOPs. Under the $1\times$ training schedule, our S3 model is also superior to other backbones, demonstrating its robustness and good transferability.

**Semantic Segmentation.** We report the mIoU score on ADE20K [56] validation set in Tab. 3. It can be seen that our S3 models consistently achieve better performance than ResNet [14], DeiT [40] and Swin [27]. Specifically, S3-T is +3.5 mIoU superior to ResNet-50. Compared to other transformer-based models, S3-T is +2.42 mIoU higher than DeiT-S, and +0.46 mIoU higher than Swin-T. S3-S outperforms ResNet-101 with a large margin (+4.46). It also surpasses DeiT-B and Swin-S by +3.63 and +0.44 mIoU, respectively. Note that we tried to add deconvolution layers to build a hierarchical structure for DeiT, but did not get performance improvements.

**Visual Question Answering.** As shown in Tab. 4, we report the accuracy on the test-dev and test-std set of VQA v2.0 dataset [54] for different vision backbones. Our model S3-T brings more than +2% accuracy gains over ResNet-101 [14], and surpassing Swin-T [27] by around 0.6% accuracy under aligned model complexity. All the above experiments demonstrate the good generality of the searched models as well as proving the effectiveness of the searched design space of vision transformer.

## 5    Related Work

**Vision Transformer**. Originally designed for natural language processing, Transformer has been recently applied to vision tasks [4, 55, 5, 46]. ViT [10] is the first one that introduces a pure transformer architecture for visual recognition. Alleviating the data hungry and computational issues, DeiT [40] introduces a series of training strategies to remedy the limitations. Some further works resort to feature pyramids to purify multi-scale features and model global information [44, 15, 31]. Swin Transformer [27] introduces non-overlapping window partitions and restricts the calculation of self-attention within each window, leading to a linear runtime complexity *w.r.t.* token numbers.

**Search Space**. There is a wide consensus that a good search space is crucial for NAS. Regarding CNN, the current search space, such as cell-based space [26, 7, 49] and MBConv-based space [3, 32, 51], is pre-defined and fixed during search. Several recent works [17, 29, 30, 33, 25, 23] perform search space shrinking to find a better compact one. Moreover, NSENet [8] proposes an evolving search space algorithm for MBConv space optimization. RegNet [35] presents a new paradigm for understanding design space and discovering design principles. However, the exploration of vision transformer space is still limited, and our work is the first along the direction.

**Search Algorithm**. There has been an increasing interest in NAS for automating network design [11, 20]. Early approaches use either reinforcement learning [57, 58] or evolution algorithms [47, 36]. These approaches require training thousands of architecture candidates from scratch. Most recent works resort to the one-shot weight sharing strategy to amortize the searching cost [22, 34, 1, 12]. The key idea is to train an over-parameterized supernet where all subnets share weights. For transformer, there are few works employing NAS to improve architectures. Evolved Transformer [37] searches the inner architecture of transformer block. HAT [43] proposes to design hardware-aware transformers with NAS to enable low-latency inference on resource-constrained hardware platforms. BossNAS [21] explores hybrid CNN-Transformers with block-wisely self-supervised. The most related work is AutoFormer [6] which provides a simple yet effective framework of vision transformer search. It could achieve once-for-all training with newly proposed weight entanglement. In this work, we focus on both the search space design and architectures of vision transformer.

## 6 Conclusion

In this work, we propose to search the search space of vision transformer. The central idea is to gradually evolve different search dimensions guided by their *E-T Error* computed using a weight-sharing supernet. We also provide analysis on vision transformer, which might be helpful for understanding and designing transformer architectures. The search models, S3, achieve superior performance to recent prevalent ViT and Swin transformer model families under aligned settings. We further show their robustness and generality on downstream tasks. In further work, we hope to investigate the usage of S3 in *CNN* search space design.

## Acknowledgements and Disclosure of Funding

Thank Hongwei Xue for providing helps and discussions on vision-language tasks. No external funding was received for this work.

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
