# Searching the Search Space of Vision Transformer
## —— Supplementary Material ——

**Minghao Chen**[1,*], **Kan Wu**[2,*], **Bolin Ni**[3,*], **Houwen Peng**[4,†],
**Bei Liu**[4], **Jianlong Fu**[4], **Hongyang Chao**[2], **Haibin Ling**[1]
[1]Stony Brook University, [2]Sun Yat-sen University,
[3]Institute of Automation, CAS, [4]Microsoft Research Asia

This supplementary material contains additional details of Section 2.4, 3 and 4.4 and a discussion about the broader impacts of this paper. The details include:

- **Searching in the searched space.** We provide the details of the two steps for vision transformer search: (1) Supernet training without resource constraints; (2) Evolution search under resource constraint.
- $Q$-$K$-$V$ **dimension could be smaller than the embedding dimension.** We suppose the underlying reasons might be that the feature maps of the different heads are similar in deeper layers. We visualize the attention maps of the heads in deep layers and find they are consistent with our assumption.
- **Experimental settings.** For Sec. 4.4 Vision and Vision-Language Downstream Tasks (in the main manuscript), we provide the experimental settings in detail.

## A    Searching in the Searched Space

In this section, we present the details of supernet training and evolutionary algorithm. Alg. 1 elaborates the procedure of supernet training with sandwich strategy. In each iteration, we sample the largest $\alpha_{max}$, smallest $\alpha_{min}$, and two random middle models $\alpha_1, \alpha_2$. Their weights are inherited from the supernet's weights $W_{\mathcal{A}}$. We compute losses using the subnets and backward the gradients. At last, we update the corresponding weights with the fused gradients.

Alg. 2 shows the evolution search in our method. For crossover, two randomly selected candidate architectures are picked from the top candidates firstly. Then we uniformly choose one block from them in each layer to generate a new architecture. For mutation, a candidate mutates its depth and embedding dimension in each stage with probability $P_d$ and $P_e$ firstly. Then it mutates each block in each layer with a probability $P_m$ to produce a new architecture. Newly produced architectures that do not satisfy the constraints will not be added to the next generation. Specifically, we set the size of population to 50 and the number of generation steps to 20. In each generation step, top 10 architectures are picked as the parents to generate child networks by mutation and crossover. The mutation probability $P_d$, $P_e$ and $P_m$ are set to 0.2, 0.2 and 0.4, respectively.

## B    $Q$-$K$-$V$ Dimension Could be Smaller Than the Embedding Dimension.

In Sec. 3 **Analysis and Discussion**, we find that $Q$-$K$-$V$ dimension could be smaller than the embedding dimension. We suppose the underlying reasons might be that the feature maps of the different heads are similar in deeper layers. To verify the assumption, we feed Fig. 4(a) (in the main manuscript) into the once-for-all supernet, and visualize the attention map in the stage 3 - block 7. As shown in Fig. 1, the attention maps in (d), (e), (h) and (j) are very similar. Besides, the ones of (b)

---

*Equal contributions. Work done when Minghao, Kan, Bolin are interns of MSRA. †Corresponding author.

35th Conference on Neural Information Processing Systems (NeurIPS 2021), Sydney, Australia.

---

**Algorithm 1** Supernet training without resource constraints

---

**Input:** Training epochs $N$, search space $\mathcal{A}$, supernet $\mathcal{N}$, initial supernet weights $W_\mathcal{A}$, train dataset $D_{\text{train}}$, loss function *Loss*

**Output:** Well-trained supernet

1: **for** $i := 1$ to $N$ **do**
2:     **for** *data, labels* in $D_{\text{train}}$ **do**
3:         Sample the largest $\alpha_{max}$, smallest $\alpha_{min}$, and two random middle models $\alpha_1, \alpha_2$ from the space $\mathcal{A}$
4:         Obtain the corresponding weights $W_{\alpha_{max}}, W_{\alpha_{min}}, W_{\alpha_1}, W_{\alpha_2}$ from $W_\mathcal{A}$
5:         Compute the gradients $\nabla W_{\alpha_{max}}, \nabla W_{\alpha_{min}}, \nabla W_{\alpha_1}, \nabla W_{\alpha_2}$ based on *Loss, data, labels*
6:         Update the corresponding parts in $W_\mathcal{A}$
7:     **end for**
8: **end for**

---

---

**Algorithm 2** Evolution search under resource constraints

---

**Input:** Search space $\mathcal{A}$, supernet $\mathcal{N}$, supernet weights $W_\mathcal{A}$, population size $P$, resources constraints $C$, number of generation iteration $\mathcal{T}$, validation dataset $D_{\text{val}}$, mutation probability of depth $P_d$, mutation probability of embedding dimension $P_e$, mutation probability of each layer $P_m$

**Output:** The most promising transformer $\alpha^*$

1: $G_{(0)} :=$ Randomly sample $P$ architectures $\{\alpha_1, \alpha_2, \cdots \alpha_P\}$ from $\mathcal{A}$ with the constrain $C$
2: **while** search step $t \in (0, \mathcal{T})$ **do**
3:     **while** $\alpha_i \in G_{(t)}$ **do**
4:         Obtain the corresponding weight $W_{\alpha_i}$ from the supernet weights $W_\mathcal{A}$
5:         Obtain the accuracy of the subnet $\mathcal{N}(\alpha_i, W_{\alpha_i})$ on $D_{\text{val}}$
6:     **end while**
7:     $G_{\text{topk}} :=$ the top $K$ candidates by accuracy order;
8:     $G_{\text{crossover}} :=$ Crossover$(G_{\text{topk}}, \mathcal{A}, C)$
9:     $G_{\text{mutation}} :=$ Mutation$(G_{\text{topk}}, P_d, P_e, P_m, \mathcal{A}, C)$
10:     $G_{(t+1)} := G_{\text{crossover}} \cup G_{\text{mutation}}$
11: **end while**
12: $\alpha^* :=$ best architecture in $G_{(\mathcal{T}+1)}$ in terms of accuracy

---

and (j) are close to each other as well. Therefore, the self-attention module does not require a large $Q$-$K$-$V$ dimension (number of heads) as the embedding dimension.

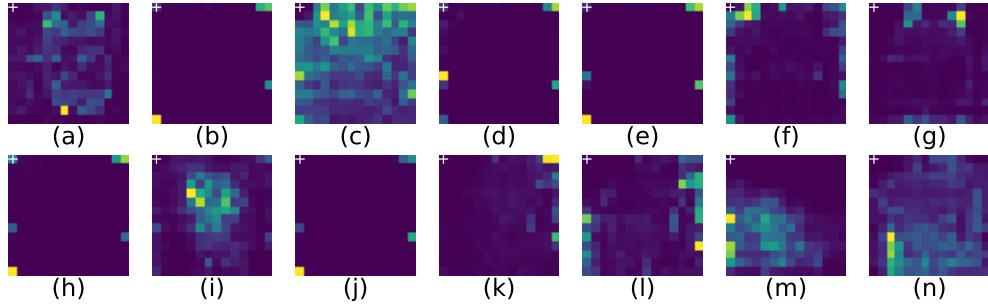

Figure 1: Attention maps of different heads in the stage 3 - block 7 of the one-step pretrained supernet. The number of heads is 14, and the reference position is on left top, marked with +.

## C Experimental settings

In this section, we provide the detailed experimental settings of our models when transferred to downstream vision and vision-language tasks, including object detection, semantic segmentation and visual question answering.

**Object Detection.** We conduct the object detection experiments on COCO dataset [7]. We train the model on training set (∼118k) and evaluate it on the validation set (5k). We replace the backbone of Cascade Mask R-CNN [1] with our discovered S3-Transformer and compare its performance with other prevalent backbones, including CNNs and handcrafted transformers, under the similar computational cost. All the models are conducted on MMDetection [2]. Same as [8], we utilize multi-scale training and 3x schedule (36 epochs). The initial learning rate is $1 \times 10^{-4}$. The optimizer is AdamW[9] with 0.05 weight decay. The experiments are conducted on 8 Tesla V100 GPUs and the batch size set to 16.

**Semantic Segmentation.** We choose ADE20K dataset [13] to test the representation power of our model on semantic segmentation task. It is a widely used scene parsing dataset which contains more than 20K scene-centric images and covers 150 semantic categories. The dataset is split into 20K images for training and 2K images for validation. For a fair comparison, we use the totally same setting as Swin [8], including the framework UperNet [11] and all the hyperparameters. The backbone is initialized with the pre-trained weights on ImageNet-1K. Specifically, since DeiT [10] adopt absolute position embeddings, we utilize bicubic interpolation to fit a larger resolution in segmentation task, before fine-tuning the model. We tried to add deconvolution layers to build a hierarchical structure for DeiT but it does not increase the performance. All the models are trained 160k iterations under MMSegmentation [3] framework on 8 Tesla V100 GPUs.

**Visual Question Answering.** We choose VQA 2.0 [12] dataset, which contains 433K train, 214K val and 453K test question-answer pairs for 204,721 COCO [7] images. We follow the framework and hyperparameters of SOHO [6] without visual dictionary module, and replace the vision backbone with the compared models. The last stage of vision backbone outputs vision tokens directly. The image resolution is $384 \times 384$. The weights of vision backbone and cross-modal transformer are initialized based on ImageNet [4] and BERT [5], respectively. We employ AdamW [9] optimizer for 40 epochs with 500 iterations warm-up, a learning rate decay by 10 times at $25^{th}$ and $35^{th}$ epochs, and batch size of 2048. The initial learning rates of vision backbone and cross-transformer are $5 \times 10^{-6}$ and $5 \times 10^{-5}$, respectively. An image is paired with four texts per batch, including two matched pairs and two unmatched pairs.

## Broader Impacts and Societal Implications

This work does not have immediate societal impact, since the algorithm is only designed for finding good vision transformer models focusing on image classification. However, it can indirectly impact the society. For example, our work may inspire the creation of new vision transformer model and applications with direct societal implications. Moreover, compared with other NAS methods that require thousands of models training from scratch, our method requires much much less computation resources, which leads to much less $CO_2$ emission.