# OpenReview forum: "Searching the Search Space of Vision Transformer"
_NeurIPS.cc/2021/Conference — NeurIPS 2021 Poster_

### Official Review · Reviewer_3mCn · 2021-07-16

**Rating:** 7
**Confidence:** 5

**Summary:**

This work propose to first search for the search space for transformers and then applies once-for-all NAS methods to search for architectures within the searched search space. In particular, the work proposes an E-T error metric to evaluate the quality of the search spaces. From the results of the whole search process, this work also summarizes some design guidelines for vision transformers.
The experiments are extensive and they show the superiority of the proposed search method.

**Limitations And Societal Impact:**

Yes, the limitations are addressed.

**Main Review:**

Pros:
- The paper is easy to follow.
- The idea of search the search space first is intuitive.
- The experiments on different downstream tasks are extensive.
- The design guidelines for transformer are good for the research community.

Cons:
- The improvements over Swin transformers on object detection and semantic segmentation are not so obvious, comparing to the gap between Swin and the other methods.
- What is the search cost (GPU-days) of each step?
- The ablation study seems not enough. Are there any ablations on the E-T error, such as changing the weighting of E and T, or varying the number of sampled architectures for computing E and for computing T.

**Time Spent Reviewing:**

1.5

---

> ### Author Response · Authors · 2021-08-10
> **Response to reviewer 3mCn**
>
> Thank you very much for constructive suggestions and valuable feedback! Answers for specific questions are provided as follows, and we will integrate them in the revision.
>
> >**Q**: Improvement on object detection and semantic segmentation tasks.
>
> **A**: We hypothesize that the large improvement of Swin Transformer on object detection and semantic segmentation over other methods is due to the high capacity of the vision transformer and its hierarchical architecture design, where S3 models share a similar design in these two aspects. Besides, Swin Transformer is dedicated designed by experts with prior knowledge. Lastly, S3 models achieve a clear improvement over Swin Transformer under similar complexity. In specific, S3 has around 0.3 mAP - 0.5 mAP and 0.4\% improvements over Swin Transformer on object detection and semantic segmentation tasks.
>
> >**Q**: Search cost.
>
> **A**: The supernet training process of each search space evolution cost around 2.5 days using 16 Tesla V100 GPUs, and calculating the E-T error cost around 0.4 days in the same setting. The overall cost of S3 is about 9 days in the setting.
>
> >**Q**: Ablation study of the E-T error.
>
> **A**: Our S3 algorithm is not sensitive to the coefficient of the E-T error. According to our experiments, a relatively balanced coefficient, such as 0.3 and 0.7, will result in similar searched space. However, if we remove either the E part or the T part, the performance of searched architectures will have a clear drop of around 0.6\%. We will add more ablation study of the E-T error to the revision including the sensitivity of coefficient and the number of architectures in the calculation of E-T error.

---

### Official Review · Reviewer_YJYN · 2021-07-16

**Rating:** 6
**Confidence:** 4

**Summary:**

This paper proposes a method for searching the search space of vision. The authors propose decomposing the search space by the search dimensions and evaluating them with the proposed E-T error to explore the optimal search space. Then the new subspace is formulated as Equation (8). The results reported in this paper have achieved state-of-the-art results.

**Limitations And Societal Impact:**

Yes.

**Main Review:**

(+) This paper is easy to follow.

(+) State-of-the-art performance. This paper conducts extensive experiments and achieves impressive performance improvements.

(+) Searching the search space is a good problem in NAS methods.

(+) This paper provides lots of interesting and useful observations in section 3, which may help the development of the research of vision transformer.

(-) The proposed method seems very time-consuming in searching search space. Simply calculating each search space's expected error for all the search space is computationally inefficient (as in the 5 line in Algorithm 1). I wonder how many search spaces are considered in the searching process.

(-) I am not satisfied with the Experimental part. The authors did not provide the final searched search space nor the searched architecture. If the authors can provide detailed evidence about this issue, I will support accepting this paper. I may change my rate according to the rebuttal.


**Time Spent Reviewing:**

0.75

---

> ### Author Response · Authors · 2021-08-10
> **Response to reviewer YJYN**
>
> Thank you for the constructive suggestions and valuable feedback! Answers for specific questions are provided as follows:
>
> >**Q**: Efficiency of computing the E-T error.
>
> **A**: To effectively calculate the E-T error, we propose to use the supernet to estimate the performances of different architectures. This approach largely saves the computation of the E-T error. For each search space evolution iteration, about 60 spaces are explored (different stages and search dimensions have their own search spaces). The total time of calculating the E-T error cost of all spaces is around 0.4 days with 16 Tesla V100 GPUs. We will clarify this in revision.
>
> >**Q**: Final search space and architectures.
>
> **A**: Figure 2 shows the whole search space evolution process and reveals the final search space. Specifically, the third column of each figure indicates the final search space. For example, according to the third figure (represent the third stage) in the first row (represent the sub search space of blocks number), the final search space of the number of blocks in stage 3 is {6,7,8}.  As for the searched architectures, thanks for your suggestion, we will add them in the revision (you can find the detailed architectures [here](https://drive.google.com/drive/folders/1Ty-7fUehMrj9jfJGi85YjmCjLsOQWuh7?usp=sharing)).

---

### Official Review · Reviewer_W7fM · 2021-07-19

**Rating:** 5
**Confidence:** 5

**Summary:**

This paper introduces a neural architecture search method called S3 that first searches the search space before identifying good architectures from the discovered search space.  The authors focus on applying S3 to vision transformers and design a search space modeled after ViT.  They evaluate S3 discovered architectures on various computer vision tasks and show good performance on image classification, object detection, and semantic segmentation.


**Limitations And Societal Impact:**

The authors do not adequately evaluate their proposed S3 method for sensitivity to internal parameters and robustness/transferability to other search spaces.

**Main Review:**

## Originality
The main points of novelty introduced in this paper are:
- Applying one-shot NAS approaches to vision transformers. The authors note that in contrast to Once-for-all and BigNAS style approaches for CNNs, inplace distillation did not help when training the vision transformer supernetwork.
- Defining the search space for vision transformers.
- Searching the search space using the E-T Error heuristic along with a linear parameterization to guide the search.
The novelty though is fairly small in scope so I lean towards a neutral rating here.  Additionally, there is a point to be made that the search space should be as large as possible and include all possible sub-spaces of interest.  In this case, the search algorithm would be responsible for efficiently identifying the most suitable architectures from the space.

## Quality
The E-T error and linear parameterization are not well motivated and I want to see further justification for choosing these heuristics.

I liked the discussion provided in Section 3 of the search phase results but the writing con be improved significantly.

The empirical results in Section 4 are promising and demonstrate S3 to yield highly performant architectures.

However, I have quite a few questions that I hope the authors can address:
- What do you mean by compose the evolved search space in Figure 1?
- It appears that the relationships between E-T error and a search dimension is not always linear.  Have you observed failure cases where using a linear parameterization didn't learn a good search space?
- How sensitive is the search process to the parameter \tau?  Have you tried other values?
- What is the search cost for 3 iterations?  Have you conducted search space refinement to convergence?
- Have you tried comparing S3 to a one-shot model trained on a larger search space that includes the resulting search spaces from all iterations of the search process?
- Have you tried extending the search space refinement process to other NAS benchmarks like the popular MobileNetV3 search space?  Does S3 transfer to these other search spaces?

## Clarity
The paper is poorly written and errors are littered across the entire text.  I have enumerated a subset of issues I noticed but there are certainly more.

_Need clarification_
- L45: "the third stage is the most important..." what is the third stage? this is not introduced until much later
- There are 4 steps in Figure 1 but there is nothing corresponding to these four steps in the actual text.  I see steps 1 and 2 in section 2.3 but not 3 and 4.

_Typos & Grammar_
- L37-38: "Different from [34] that trains..." -> In contrast to [34], which trains...
- L40-41: "We decouple the search space by multiple dimensions..." -> We characterize search spaces by multiple dimensions...
- L42-43: "Particularly, we use linear parameterization to model the tendencies of different dimensions so as to guide the search." -> In particular, we parameterize the model along different search dimensions...
- L44: "we analyze the search process of search space and provide some interesting observations" -> needs to be rephrased
- L49: "we hope these observations could help..." -> we hope these observations will help
- L:104: "Different form RegNet [34] that trains..." -> needs to be rephrased
- Figure 2 caption cut off
- L145: "Fig. 3 shows an direct..." -> Fig. 3 shows a direct...
- L147: "The third stage is the most important and increasing blocks of it contribute to better performance" -> needs to be rephrased
- L151-153: needs to be rephrased
- L153: "should include less layers" -> should include fewer layers
- L154: use a different verb from prefer perhaps "more compatible" or "more suitable"
- L156: "deep layers attend to" -> deep layers tend to
- L157: "We assume that transformers increase focusing region with deeper layers" -> needs to be rephrased
- L161: "which illustrated from" -> which is illustrated from
- "focusing regions" -> regions of focus
- L164: "Deeper layer, larger MLP ratio" -> phrase as a sentence to mirror other points.
- L166: "It gives our insight to design different MLP ratio" -> needs to be rephrased
- L167: "Q-K-V dimension could smaller than the embedding dimension" -> Q-K-V dimension can be smaller than the embedding dimension
- L171-173: needs to be rephrased
- L172: "less heads" -> fewer heads
- L215: "we could get model..." -> we found a model...

## Significance
Despite the impressive empirical results, the significance of the paper is greatly limited by the numerous grammatical errors that litter the text. I would evaluate its potential for impact to be low in its current state.

===============Post Author Response===============

After reading the authors' response, I have decided to change my score to a 5.  I am hesitant to recommend acceptance given the unpolished state of the paper at submission.  I do see the merit of the method proposed and believe it would benefit tremendously from improved  writing and additional experiments evaluating S3 on different search spaces in a future submission.

**Time Spent Reviewing:**

3

---

> ### Author Response · Authors · 2021-08-10
> **Response to reviewer W7fM**
>
> Thank you for constructive suggestions and valuable feedback! Responses for specific questions are provided as follows, and we will integrate them in future revision.
>
> > **Q**: Novelty.
>
> **A**: Thanks for your valuable comments. Here we would like to highlight our key novelty: First, to the best of our knowledge, we are the first to automatically search for a good search space of vision transformer models. Specifically, we use the E-T Error heuristic along with a linear parameterization to guide the space search. Search space is crucial because it determines the performance bounds of architectures during the search, and the search space design of vision transformer is barely explored and challenging. Second, we provide design guidelines and analysis of vision transformer models based on the search space evolution process. These guidelines could be directly applied to vision transformer architectures. For example, they improve Swin-T by 0.6% top-1 accuracy on ImageNet with comparable FLOPs and parameters. Third, we propose a simple and effective pipeline for vision transformer search with minimal human interactions. Extensive experiments demonstrate that our method yield high-performing architectures.
>
> > **Q**: Linear parameterization.
>
> **A**: There are two main reasons why we use linear parameterization to approximate the E-T error. First, since a relatively small range of the search dimension is considered in each iteration, the coefficient can be viewed as an approximation of the derivative (first-order Taylor approximation), which is used to indicate the evolution tendency of the search space. Second, for each evolution iteration, we only have at most three points for each search dimension. Linear parameterization is therefore adopted and we empirically find it works very well and yields good search space.
>
> > **Q**: Meaning of composing the search space.
>
> **A**: In our algorithm, we first decompose the original search space into different search dimensions such as MLP ratio, number of blocks, and window size. Then, each search dimension is evolved independently. Therefore, we use the word "compose" to indicate that the evolved search dimensions together form the evolved search space. We will clarify this in revision.
>
> > **Q**: Failure cases where using a linear parameterization does not learn a good search space.
>
> **A**: Yes, the relationship between E-T error and a search dimension may not be rigorously linear. However, the linear parameterization effectively and efficiently predicts the tendency of the evolution of the search space. Thus motivated, in our experiment, we iteratively improve the search space using linear parameterization.
>
> > **Q**: Hyperparameter $\tau$ and search iteration.
> >
> **A**: The value $\tau$ is picked according to our experiments. We find that a larger $\tau$ makes the search space evolution process converge more slowly and hence costs more GPU time. To balance the convergence speed (GPU time cost) and final performance, we found 0.4 is appropriate, using which the search space evolution process almost converges at the third iteration under this setting. The total search cost of 3 iterations is about 9 days using16 Tesla V100 GPUs.
>
> > **Q**: Compare with one-shot methods on a large space.
>
> **A**: Ideally, the search space should include all the possible architectures. However, it is impractical as commonly agreed in many previous studies. As illustrated in [1,2,3], effectively searching over a large search space can be very hard and time-consuming. We tried to search over a large space but found that the corresponding supernet converged badly with the same recipe as S3. Besides, the key idea of S3 is to search the search space of vision transformer instead of merely searching for specific architectures, thus the search space found through all iterations is part of the algorithm results. The space searched by S3 might serve as a good foundation for further vision transformer architecture search studies.
>
> > **Q**: Apply S3 to other search spaces.
>
> **A**: Thanks for the suggestion. Conceptually, S3 might be applicable to CNN search space design. In this work, we mainly focus on vision transformer search space design which is a more challenging problem compared with the well-established CNN search space. We expect our works to shed light on vision transformer search space design and architecture design. In the future, we will explore other search spaces (e.g., MobileNetV3) as suggested.
>
> > **Q**: Clarity.
>
> **A**: Sorry for the confusion. We will add an additional explanation for the "third stage" in line 45 to the revision. As for the four steps in Figure 1, the first three steps correspond to Equations (7), (8), (9) respectively, and the fourth step corresponds to L10 in Algorithm 1. We will make it clearer in the revision.
>
> >**Q**: Typos and grammar errors.
>
> **A**: Thank you very much for the detailed comments, we will carefully polish the draft to fix these issues and to improve the writing quality in the revision. We agree that there are rooms to improve in terms of writing, we feel that these issues should not be a main obstacle against acceptance of the work, since they are fixable and will be fixed. We truly appreciate your suggestions and sincerely hope this response can be helpful in the final evaluation of the paper.
>
> [1] Xiang Li, Chen Lin, Chuming Li, Ming Sun, Wei Wu, Junjie Yan, and Wanli Ouyang. Improving one-shot nas by suppressing the posterior fading. In CVPR, 2020.
>
> [2] Niv Nayman, Asaf Noy, Tal Ridnik, Itamar Friedman, Rong Jin, and Lihi Zelnik. Xnas: Neural architecture search with expert advice. In NeurIPS, 2019.
>
> [3] Asaf Noy, Niv Nayman, Tal Ridnik, Nadav Zamir, Sivan Doveh, Itamar Friedman, Raja Giryes, and Lihi Zelnik. Asap: Architecture search, anneal and prune. In AISTATS, 2020.

---

### Official Review · Reviewer_uQyW · 2021-07-19

**Rating:** 6
**Confidence:** 4

**Summary:**

This work applies NAS to the vision transformer architecture. Besides searching the architectures, it also automatically adjusts the search space guided by the E-T error that combines the error distribution and top model errors to evaluate the quality of different choices in a search dimension. By fitting a linear function to approximate the E-T error, the new search space is updated through a simple formula. They used the supernet to efficiently compute the E-T errors, and use evolutionary search to search for the best algorithm in a given search space. The discovered architecture is shown to be competitive to the state-of-the-art models, and also generalize to other vision tasks.


**Limitations And Societal Impact:**

I listed a few additional analysis in the "Weakness" section in the Main Review that would help shed more light on the limitations of the proposed methods.

**Main Review:**

Strength:

(1) The idea of automatically iterating the search space and using super-net to make the calculation of the E-T error are interesting. It combines existing ideas in designing the search space and weight sharing NAS and proposes a new method to update the search space by fitting a linear function to the E-T error.

(2) The evaluation result is competitive and additional experiments are conducted on several vision tasks to verify the discovered architectures' generality.

(3) Analysis are performed on how the search space changes in different iterations, however I had some questions regarding the reliability of the results.

(4) The paper is well written and easy to follow. (A minor issue in formatting, the caption of Figure 2 is covered a bit by Figure 3.)

Weakness:

(1) This paper combines several existing ideas and applies them to a new domain (Vision Transformer), and the novel contribution in the method is the way to update the search space using the linear function fitting of the search choices and the E-T error. However, there should be more analysis or supporting evidence for this simplifying assumption in the experiments. For example, it would help to add some plots and calculate the R-squared to verify how good this linear approximation assumption is.

(2) Some of the hyperparameter choices need more discussion. For example, how is the value of tau (0.4) picked? Was it tuned over some range? How sensitive is the result with respect to this value? Same for the number of search space iterations. Why does it stop at 3, is it due to compute limits or maybe 3 is the sweet spot? It would help to add some analysis regarding how the accuracy changes with the number of search space iterations.

(3) The analysis of the trends in Section 3 is quite interesting. Since the conclusions are draw from the results in Figure 2, I wonder how much variance is in the result. For example, if you repeat the experiments multiple times, how much would the search space evolution differ between different repeats. This would help evaluate the reliability of the analysis.




**Time Spent Reviewing:**

3

---

> ### Author Response · Authors · 2021-08-10
> **Response to reviewer uQyW**
>
> Thank you for the constructive suggestions and valuable feedback! Answers for specific questions are provided as follows:
>
> > **Q**: Linear assumption of E-T Error.
>
> **A**: There are two main reasons why we use linear parameterization to approximate the E-T error. First, since a relatively small range of the search dimension is considered in each iteration, the coefficient can be viewed as an approximation of the derivative (first-order Taylor approximation), which is used to indicate the evolution tendency of the search space. Second, for each evolution iteration, we only have at most three points for each search dimension. Linear parameterization is therefore adopted and we empirically find it works very well and yields good search space. To further verify the assumption, as suggested, we plot the E-T error of different values of different search dimensions. We find the R-squared is relatively high. For example, the R-squared of using linear parameterization for different values (three points in total) of the block numbers in stage 1 is 0.97. We will add the plot and more details to the revision.
>
>
> > **Q**: The choices of hyperparameters and search iteration.
>
> **A**: The value $\tau$ is picked according to our experiments. We find that a larger $\tau$ slows down convergence of the search space evolution and hence costs more GPU time. To balance the convergence speed (GPU time cost) and final performance, we found 0.4 is appropriate, using which the search space evolution process almost converges at the third iteration under this setting.  Figure 6 shows the empirical error distribution function of each iteration, illustrating the search space gets improved. In addition, we also provided the performance of searched architectures of each iteration in Table 5.
>
>
> > **Q**: Reliability of the analysis in Section 3
>
> **A**: The search space evolution process will be stable if we fix the training recipe and hyperparameters. This is because that the E-T error is not only calculated by a single architecture but one hundred representative architectures covered in the search space, which stabilize the evolution process. In addition, we directly applied our design principles to the Swin Transformer models. We found they improved Swin-T by 0.6\% top-1 accuracy on ImageNet with comparable FLOPs and parameters.
>
> We will include the above clarification, results and analysis in the future revision.

---

### Decision · Program_Chairs · 2021-09-27

**Decision:**

Accept (Poster)

**Comment:**

Four experts reviewed this paper and gave ratings 6, 6, 6, and 5, respectively --- Reviewer 3mCn decided to change 7 to 6 in a private discussion. The reviewers expressed concerns about novelty and writing. Particularly, some reviewers commented that the E-T error heuristic and linear approximation were not well-motivated, and the novelty was small. Reviewer W7fM felt strongly about the “poor state of the writing.” However, the reviewers generally appreciated the study of NAS for Transformers, and a reviewer considered the design of the Transformer-tailored search space original. AC agreed that such a study could benefit future research on Transformers and felt that the concerns can be addressed in a reasonable revision of the paper. Hence, the decision is to recommend the paper for acceptance. The authors are encouraged to make necessary changes in the camera-ready to address the reviewers' questions to the best of their ability. Especially, the authors may consider finding a native English speaker to help polish the writing. We congratulate the authors on the acceptance of their paper!